# DEEPFAKE DETECTION WITH CONTRASTIVE LEARNING IN CURVED SPACES

## ABSTRACT

Deepfake detectors excel in familiar scenarios but falter when faced with new generation techniques. Improving their generalization can be achieved through synthetic data during training or one-class anomaly detectors. However, existing techniques, limited to non-negative-curvature spaces, struggle to effectively identify counterfeit features on the intricate and diverse non-Euclidean human face manifold. Human faces defy simple Euclidean geometry due to their complexity. To overcome this limitation, we introduce a novel and efficient deepfake detector, called CTru, that learns a rich representation of facial geometry across multiple-curvature spaces in a self-supervised manner. During inference, the fakeness score is computed by integrating angle-based similarity in spherical space and model confidence in hyperbolic space with Busemann distance. CTru establishes new SoTA results on various challenging datasets in both cross-dataset and cross-manipulation scenarios, while being trained only on pristine faces, highlighting its impressive generalization performance. Code source will be made available.

## 1 INTRODUCTION

Cutting-edge improvements in deep generative models, like GANs (Goodfellow et al., 2014), diffusion models (Ho et al., 2020), and normalizing flows (Dinh et al., 2017), have enabled the creation of highly authentic counterfeit images. This has given rise to malevolent alterations of human faces, or "deepfakes", which constitute a significant menace to both individuals and society (Oltermann, 2022; Chowdhury & Lubna, 2020). In response, efforts are underway to develop tools to detect these forgeries. Existing detectors (Masi et al., 2020; Chollet, 2017; Dang et al., 2020; Rössler et al., 2019; Hsu et al., 2020) currently rely on datasets containing both authentic and fake faces to learn a binary decision boundary (Li et al., 2020b; Jiang et al., 2020; Korshunov & Marcel, 2018; Dolhansky et al., 2019; 2020; Rössler et al., 2019; Yang et al., 2019). While effective on familiar deepfake types, these detectors struggle to generalize when confronted with new facial manipulations, limiting their practical utility (Zi et al., 2021).

One effective approach to improve the generalization of deepfake detectors is to incorporate synthetic data (or *pseudo-fakes*) during training, which encourages the models to learn more adaptable decision boundaries. Many state-of-the-art (SoTA) detectors (Zhao et al., 2021; Shiohara & Yamasaki, 2022; Chen et al., 2022a; Li et al., 2020a; Chen et al., 2022b) follow this strategy by either generating additional fake images to enrich the diversity of available deepfakes or relying exclusively on synthetic deepfakes for training the detection models. However, those classification-based approaches are prone to over-fitting on training data, leading to the exploration of alternative approaches. One-class anomaly detection (OC-AD) techniques provide a promising solution to identifying anomalies/fakes in data that deviate from expected patterns (real faces) (Feng et al., 2023; Larue et al., 2023). However, in these methods, the feature representation learned in non-negative-curvature space and the Euclidean-based distances appear sub-optimal for faces. Unlike the assumptions of a Euclidean manifold, which presumes a flat and linear space, the intricate shapes and curvatures of facial features such as the nose, eyes, and mouth defy adequate description through Euclidean geometry alone. The human face is not a rigid structure; facial expressions are non-linear deformations of facial features (e.g., when a person smiles, the shape of the face undergoes complex transformations). In other words, human faces challenge Euclidean geometry with their dynamic and non-linear features.

In this paper, we present a novel approach called named CTru ("ConTrastive learning in opposite-curvature space"), a self-supervised AD method that leverages the power of hypersphere and hyperbolic geometry to learn a rich representation in spaces with different curvatures. By training in hyperbolic space, as demonstrated in Khrulkov et al. (2020), semantic similarities and hierarchical

relationships between images can be captured effectively. As a result, our features convey richer information compared to detectors relying solely on loss in hypersphere space. Hyperbolic space offers a more flexible and expressive geometry, making it better suited for handling complex facial structures. To our best knowledge, we are the first to propose a deepfake detector that learns a representation in spaces with different curvatures.

Our contributions can be summarized as follows: 1) We propose a novel deepfake detector named CTru, designed to learn a representation across multiple-curvature spaces, with an enhanced capability to model the image. 2) Our multi-objective learning method uses complementary losses to extract high-level semantic features from images during training. We then leverage these features to develop efficient "fakeness" scores based on the model confidence in hyperbolic space and cosine similarity in hypersphere space. 3) CTru establishes new SoTA results on various challenging datasets in both cross-dataset and cross-manipulation scenarios, while using only pristine faces during training. CTru not only outperforms its counterparts in terms of higher accuracy and faster inference speed but also demonstrates superior versatility, as it does not require prior knowledge of the face. In other words, it is a more robust and generic detector.

## 2 RELATED WORK

**Deepfake detection.** Numerous deepfake detectors have been proposed in the literature (Tolosana et al., 2020; Aneja & Nießner, 2020; He et al., 2021; Shao et al., 2022; Zhihao et al., 2022; Dong et al., 2022). Early detectors relied on detecting the visual artifacts of deepfake-generation techniques (Amerini et al., 2019; Li & Lyu, 2018; Afchar et al., 2018; Das et al., 2021; Guo et al., 2022), while others used frequency-domain representations to discriminate between real and fake images (Rössler et al., 2019; Liu et al., 2021; Qian et al., 2020; Fei et al., 2022). Some models utilized temporal features for deepfake detection (Guan et al., 2022; Feng et al., 2023; Zheng et al., 2021; Zhuang et al., 2022), while others focused on continual learning (Kim et al., 2021; Li et al., 2023) to avoid catastrophic forgetting across different types of fakes. Recent approaches to generalize across different generation techniques include using dedicated augmentation techniques to synthesize forged images and then training a binary classification model, such as Face-Xray (Li et al., 2020a), self-blended images (SBI) (Shiohara & Yamasaki, 2022), SLADD (Chen et al., 2022a), PCL (Zhao et al., 2021), and OST (Chen et al., 2022b).

Those existing classification-based approaches frequently encounter issues with overfitting to the training data, prompting the exploration of alternative strategies. One promising solution is to use OC-AD techniques. While plenty of works detect deepfakes via a binary classifier, very few reframe the problem as self-supervised OC-AD. Feng et al. (2023) demonstrated the effectiveness of AD in detecting deepfakes. Larue et al. (2023) used an OC-AD approach to detect deepfakes.

**Hyperbolic geometry** has become popular for their powerful geometrical representations with minimal distortion (Balazevic et al., 2019; Keller-Ressel & Nargang, 2020). Different alternatives have been proposed for various layers and components in deep neural networks (Ho et al., 2020; Dinh et al., 2017). Recent works have explored prototype-based approaches with hyperbolic output spaces, including few-shot learning and zero-shot recognition Atigh et al. (2021).

**Hierarchical learning in hyperbolic spaces.** Distinguished by its constant negative curvature, hyperbolic geometry offers a valuable perspective for analyzing semantic similarities and hierarchical relationships among high-dimensional data. (Nickel & Kiela, 2017; Liu et al., 2019; Khrulkov et al., 2020; Peng et al., 2021) advocate for the use of hyperbolic spaces as more suitable alternatives to standard Euclidean spaces for representing hierarchical structures. Additionally, they excel in depicting tree-like structures and taxonomies (Ganea et al., 2018; Long et al., 2020; Liu et al., 2020).

## 3 PROPOSED METHOD

A constant curvature space is a smooth Riemannian manifold. The curvature sign dictates the space type. A negative curvature represents a hyperbolic space, and we use the Poincare ball model in this work. A zero curvature implies the Euclidean space, whereas a positive curvature denotes a hypersphere space. Our CTru (Fig. 1) learns the generic features in spaces of different curvatures.

**Notation.** For the sake of notation simplicity and without loss of generality, in the follows, we denote $\mathbf{x}$ and $\mathbf{y}$ as any two points in a space of positive or negative curvature, while the subscript $_E$ extends to Euclidean (i.e., $\mathbf{x}_E$ is a point in Euclidean space). For the sake of detail, in Section 3.3, $\mathbf{x}$ and $\mathbf{y}$ belong to $\mathbb{S}^d$ (hypershpere space), while in Section 3.4, $\mathbf{x}$ and $\mathbf{y}$ belong to $\mathbb{H}^d$ (hyperbolic space).

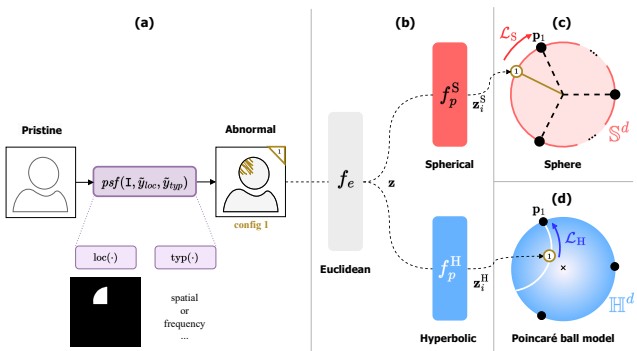

Figure 1: **Overview of our method.** (a) Given a pristine face, its abnormal version is first generated. (b) Using only one encoder $f_e$, an embedding $\mathbf{z}$ is obtained and then projected to the hypersphere $\mathbb{S}^d$ and hyperbolic $\mathbb{H}^d$ spaces using two non-parameter functions $f_p^S$ and $f_p^H$. (c)-(d) Resulting vectors are attracted to their class prototypes using $\mathcal{L}_{\mathrm{S}}$ and $\mathcal{L}_{\mathrm{H}}$.

## 3.1 OVERVIEW OF CTRU

We are given a training batch $\{(\mathbf{I}_i)\}_{i=1}^B$ with $B$ examples of pristine faces. We generate a batch of abnormal samples $\{(\mathbf{I}_i', \tilde{y}_i)\}_{i=1}^B$ by applying a data augmentation $\mathbf{I}_i' = \mathrm{psf}\,(\mathbf{I}_i, \tilde{y}_{loc}, \tilde{y}_{typ})$ where $\tilde{y}_i \in [1 \mathinner{.\,.} C]$ and $C = N_{loc} \times N_{typ}$ is the number of different anomaly configurations (Section 3.2). CTru is composed of: (i) an encoder $f_e(\theta)$ that maps an abnormal face $\mathbf{I}_i'$ to a feature vector (embedding) $\mathbf{z}_i \in \mathbb{R}^d$; (ii) two non-parameter projectors, $f_p^{\mathsf{S}}$ and $f_p^{\mathsf{H}}$, which respectively $L_2$ normalizes $\mathbf{z}_i$ to produce a vector in a spherical space $\mathbb{S}^d$, and applies the exponential map to $\mathbf{z}_i$ to produce a vector in an hyperbolic space $\mathbb{H}^d$.

Briefly speaking, CTru learns by projecting the inputs onto the $\mathbb{S}^d$ and $\mathbb{H}^d$ manifolds and attracting the projected embeddings to their self-supervised class labels, which are represented by predefined prototypes $\mathcal{P} = \{\mathbf{p}_1, .., \mathbf{p}_C\}$. CTru uses two losses, $\mathcal{L}_{\mathrm{S}}$ and $\mathcal{L}_{\mathrm{H}}$, and the global loss is

$$\mathcal{L}_{\mathrm{CTru}}\ \left(\theta, \mathcal{P}, \{(\mathbf{z}_i, \tilde{y}_i)\}_{i=1}^B\right) = \mathcal{L}_{\mathrm{S}}\left(\theta, \mathcal{P}, \{(f_p^{\mathsf{S}}(\mathbf{z}_i), \tilde{y}_i)\}_{i=1}^B\right) + \lambda \cdot \mathcal{L}_{\mathrm{H}}\left(\theta, \mathcal{P}, \{(f_p^{\mathsf{H}}(\mathbf{z}_i), \tilde{y}_i)\}_{i=1}^B\right) \quad (1)$$

where $\lambda > 0$ is the balancing hyper-parameter, that is 1 in all of our experiments for simplicity.

**How to generate the predefined evenly distributed prototypes (used for spherical and hyperbolic spaces)?** A three-step method is proposed: (1) First, we initialize the $C$ prototypes $\mathbf{p}_i$ as evenly distributed class centroids in the spherical space $\mathbb{S}^d$, leveraging the vertices of the $C$-simplex ($C \leq d+1$), as in Lange & Wu (2008). (2) Next, we add zero-dimensional vectors to the prototypes to embed them in $\mathbb{R}^d$, ensuring they are $d$-dimensional. (3) Finally, we use the Gram-Schmidt process to compute a random basis of $\mathbb{R}^d$ and project the prototypes $\mathbf{p}_i$ onto this basis to obtain final prototypes.

## 3.2 GENERATING ABNORMAL REGIONS IN THE PRISTINE FACE

Our approach involves generating highly realistic and diverse abnormal face versions by applying a rich augmention, denoted as psf(.), to a real face $\mathbf{I} \in [0, 255]^{W \times H \times 3}$. It consists of selecting from a 2D grid embedding the facial region a square patch at a chosen location $\tilde{y}_{loc} \in [1 \mathinner{.\,.} N_{loc}]$, applying a specific type of alteration $\tilde{y}_{typ} \in [1 \mathinner{.\,.} N_{typ}]$, and utilizing the blending augmentation, which is proved efficient in forgery detection (Li et al., 2020a; Chen et al., 2022a; Shiohara & Yamasaki, 2022), to generate the final abnormal face. Formally, the augmentation can be written as:

$$\mathrm{psf}\,(\mathbf{I}, \tilde{y}_{loc}, \tilde{y}_{typ}) = \underbrace{\mathbf{I} \odot \mathrm{loc}(\mathbf{I}, \tilde{y}_{loc})}_{\text{target}} + \underbrace{\mathrm{typ}(\mathbf{I}, \tilde{y}_{typ}) \odot [\mathbf{1} - \mathrm{loc}(\mathbf{I}, \tilde{y}_{loc})]}_{\text{source}} \quad (2)$$

where $\odot$ is the element-wise multiplication, $\mathrm{loc}(.)$ returns the binary mask of the $\tilde{y}_{loc}$-th patch of interest, $\mathrm{typ}(.)$ alters an image with the $\tilde{y}_{typ}$-th abnormal type, and $\mathbf{1}$ is an image with only '1's.

**The position of forged patch.** We employ the grid scheme as in Larue et al. (2023) to generate masks by dividing the face into a grid of $N_{loc} = N_{rows} \times N_{cols}$ cells. The landmarks extracted using a facial landmark detector are utilized to align the face and crop it into a bounding rectangle. The face portion is then divided into a grid of cells and a mask is generated for each cell at position $\tilde{y}_{loc}$. **The type of the anomaly.** To generate different types of abnormal regions, commonly used image

augmentation techniques are used. We categorized the augmentation techniques based on the type of the anomaly: (1) anomalies in the frequency domain, and (2) in the spatial domain. The first category includes downscaling followed by upscaling and JPEG compression, while the second category includes Gaussian noise, color jittering, and blurring. In other words, $N_{typ} = 2$.

### 3.3 LEARNING IN POSITIVELY CURVED SPACE

We define a sphere model with a positive curvature of 1 as $\mathbb{S}_r^d = \left\{\mathbf{x}_E \in \mathbb{R}^{d+1} : \|\mathbf{x}_E\| = r, r > 0\right\}$, where $r$ represents the radius. Our model works with $L_2$ normalized features, $\mathrm{f}_p^\mathrm{S}(\mathbf{z}_i) = \mathbf{z}_i/\|\mathbf{z}_i\|_2$, meaning that $r = 1$. Therefore, in the rest of the paper, we simply denote the space $\mathbb{S}^d$. The distance between two points is determined by the shortest path between them on the manifold, also known as the geodesic distance. In this space, the geodesic is the arc connecting the two points. The resulting function is called the induced geodesic distance function, defined as: $\mathrm{dist}_\mathbb{S}(\mathbf{x}, \mathbf{y}) = r \times \cos^{-1}\left(\langle\mathbf{x}, \mathbf{y}\rangle\, r^{-2}\right)$. This space is bounded, meaning that the distance between any two points cannot be arbitrarily large: $0 \le \mathrm{dist}_\mathbb{S}(\mathbf{x}, \mathbf{y}) \le r\pi$ ($r = 1$ in our model).

When working in this space, we inspire by the supervised contrastive learning (Khosla et al., 2020) (SCL) which is proven to be more efficient than the cross entropy loss. SCL involves two stages: (1) jointly training the encoder $\mathrm{f}_e$ and projection head, denoted as $\mathrm{f}_p$, to solve the task; (2) replacing $\mathrm{f}_p$ with a linear layer and training this layer to solve the classification task with the frozen representation $\mathrm{f}_e$. To further improve the feature learnt while simplifying the training step by merging both phases, the bounded contrastive regression loss is proposed (Larue et al., 2023). This loss employs a set of *predefined* prototypes positioned on the sphere which act as class centers. This loss biases the supervised contrastive loss, resulting in embeddings from the same class being pulled closer to the corresponding predefined class centers. By doing so, the loss effectively encourages embeddings of the same class to be grouped together, while simultaneously pushing apart embeddings from different classes. The loss is defined as follows:

$$\mathcal{L}_\mathrm{S}\left(\theta, \boldsymbol{\mathcal{P}}, \{(\mathbf{z}_i^\mathrm{S}, \tilde{y}_i)\}_{i=1}^B\right) = \sum_{i \in \mathcal{I}} \frac{-1}{|\mathcal{P}(i)|} \left(\log \frac{e^{\mathrm{sim}\left(\mathbf{z}_i^\mathrm{S}, \mathbf{p}_{\tilde{y}_i}\right)}}{\sum\limits_{n \in \mathcal{N}(i)} e^{\mathrm{sim}\left(\mathbf{z}_i^\mathrm{S}, \mathbf{z}_n^\mathrm{S}\right)}} + \sum_{p \in \mathcal{P}(i)} \log \frac{e^{\mathrm{sim}\left(\mathbf{z}_i^\mathrm{S}, \mathbf{z}_p^\mathrm{S}\right)}}{\sum\limits_{n \in \mathcal{N}(i)} e^{\mathrm{sim}\left(\mathbf{z}_i^\mathrm{S}, \mathbf{z}_n^\mathrm{S}\right)}}\right)$$

$$(3)$$

where $|\cdot|$ is the cardinal function, $\mathrm{sim}(\mathbf{x}_a, \mathbf{x}_b) = \langle\mathbf{x}_a, \mathbf{x}_b\rangle/(\|\mathbf{x}_a\| \cdot \|\mathbf{x}_b\|)$ is the cosine similarity, $\mathcal{I} = \{1, ..., B\}$ is the set of indices in the batch, $\mathcal{N}(i)$ is the set of indices $\{n \in [1 .. B] \mid n \ne i\}$ that forms a negative pair with the $i$-th sample in the batch, $\mathcal{P}(i)$ is the set of indices $\{p \in [1 .. B] \mid p \ne i,\ \tilde{y}_p = \tilde{y}_i\}$ that forms positive pairs with the $i$-th sample in the batch, where a positive pair consists of two samples with the same label. Our method relies on $L_2$-normalized vectors, the features have a unit norm and lie on the unit hypersphere. We use cosine similarity as the basis for the loss function, which is invariant to the magnitude of the embeddings and depends only on the angle between the two vectors. However, this approach has a drawback: the norm of embeddings may not be an indicator reliable enough for detecting fake samples during inference. We propose therefore to use a complementary objective in a hyperbolic space to learn richer features.

### 3.4 LEARNING IN NEGATIVELY CURVED SPACE

To improve the performance of our detector by capturing additional features specific to abnormal faces within a hyperbolic space, we opt to embed our data in the Poincaré ball which leverages the distinctive properties inherent to hyperbolic geometry (Atigh et al., 2021). We aim to unveil novel discriminative features that are challenging to capture within Euclidean geometry, as in the spherical space, we propel the extracted feature vectors, projected into Poincaré ball mode, towards their respective predefined prototypes (ideal points).

**Poincaré ball model** is a hyperbolic space defined as: $\mathbb{H}_r^d = \left\{\mathbf{x}_E \in \mathbb{R}^d : \|\mathbf{x}_E\|^2 < r^2, r > 0\right\}$, where $r$ represents the radius. In hyperbolic spaces, the traditional vector space formalism does not apply. A gyrovector formalism is therefore introduced (Rassias, 2010) to perform operations such as addition. In particular, for a pair of points $\mathbf{x}$ and $\mathbf{y}$ in the space, their Möbius addition is defined as: $\mathbf{x} \oplus_r \mathbf{y} = \frac{(1 + 2r^{-2}\langle\mathbf{x}, \mathbf{y}\rangle + r^{-2}\|\mathbf{y}\|^2)\mathbf{x} + (1 - r^{-2}\|\mathbf{x}\|^2)\mathbf{y}}{1 + 2r^{-2}\langle\mathbf{x}, \mathbf{y}\rangle + r^{-4}\|\mathbf{x}\|^2\|\mathbf{y}\|^2}$. A remarkable feature of hyperbolic geometry is that the length of circles and the area of disks grow exponentially, in contrast to Euclidean geometry where they grow linearly and quadratically, respectively. In this space, the induced geodesic distance is defined as: $\mathrm{dist}_\mathbb{H}(\mathbf{x}, \mathbf{y}) = 2r \times \tanh^{-1}\left(r^{-1}\|-\mathbf{x} \oplus_r \mathbf{y}\|\right)$. In hyperbolic geometry, the exponential

map $\exp_{\mathbf{x}}^r$ is a function that allows us to map a Euclidean vector to the hyperbolic manifold. More specifically, for a point $\mathbf{x}$ in the hyperbolic space $\mathbb{H}_r^d$ and a vector $\mathbf{v}_E$ in the tangent space $T_{\mathbf{x}}\mathbb{H}_r^d \cong \mathbb{R}^d$, the exponential map is defined as: $\exp_{\mathbf{x}}^r(\mathbf{v}_E) = \mathbf{x} \oplus_r \left( r \tanh \left( \frac{\|\mathbf{v}_E\| \lambda_r(\mathbf{x})}{2r} \right) \frac{\mathbf{v}_E}{\|\mathbf{v}_E\|} \right)$, where $\lambda_r(\mathbf{x}) = \frac{2}{1 - r^{-2} \|\mathbf{x}\|^2}$ is the conformal factor in Riemannian geometry.

In hyperbolic geometry, points situated at infinity are referred to as **ideal points**. These points reside on the boundary of the ball in the Poincaré model. Notably, the set of ideal points in hyperbolic space shares *identical topological properties* with the hypersphere $\mathbb{S}_r^d$, as demonstrated in (Atigh et al., 2021) (see the difference between (Atigh et al., 2021) and ours in Section 3.6). Consequently, our generated evenly-distributed *predefined points* in Section 3.1 can serve as *ideal prototypes* and can be effectively employed for embedding prototypes into hyperbolic space. Henceforth, we consider a Poincaré ball with $r = 1$ and denote it simply as $\mathbb{H}^d$ throughout the remainder of this paper.

To accurately measure the distance between embeddings and prototypes in hyperbolic space, relying on geodesic distance Section 3.4 is not a viable option. This is because every point $\mathbf{z}_i^{\mathrm{H}}$ in the Poincaré ball $\mathbb{H}^d$ is infinitely far from any prototypes $\mathbf{p} \in \mathcal{P}$ in the boundary sphere $\mathbb{S}^d$. We use therefore an alternative approach involving the **Busemann function**, which provides a normalized distance function from a point on the boundary of hyperbolic space to another point. It can be defined as:

$$\mathrm{bus}\left(\mathbf{z}_i^{\mathrm{H}}, \mathbf{p}\right) = \lim_{t \to \infty} \left\{ \mathrm{dist}_{\mathbb{H}}(\gamma_{\mathbf{p}}(t), \mathbf{z}_i^{\mathrm{H}}) - t \right\} \tag{4}$$

where $\gamma_{\mathbf{p}}(t)$ is the geodesic at $\mathbf{p}$ and $\mathbf{z}_i^{\mathrm{H}} = \mathrm{f}_p^{\mathrm{H}}(\mathbf{z}_i) = \exp_{\mathbf{0}}(\mathbf{z}_i)$. The Busemann function can be analytically solved in $\mathbb{H}^d$ and have the following closed form:

$$\mathrm{bus}_{\mathbb{H}}(\mathbf{z}_i^{\mathrm{H}}, \mathbf{p}) = \log \left( \frac{\|\mathbf{p} - \mathbf{z}_i^{\mathrm{H}}\|^2}{1 - \|\mathbf{z}_i^{\mathrm{H}}\|^2} \right) \tag{5}$$

Our loss function in negatively curved space is defined as the sum of Busemann functions with respect to each embedding and its corresponding class prototype:

$$\mathcal{L}_{\mathrm{H}}\left(\theta, \mathcal{P}, \{(\mathbf{z}_i^{\mathrm{H}}, \tilde{y}_i)\}_{i=1}^B\right) = \sum_{i \in [1..B]} \mathrm{bus}_{\mathbb{H}}(\mathbf{z}_i^{\mathrm{H}}, \mathbf{p}_{\tilde{y}_i}) \tag{6}$$

By minimizing Eq. (6), we bring the embeddings closer to the class prototypes, which correspond to the ideal points at the boundary of the Poincaré ball. In Atigh et al. (2021), a regularization term was also employed to address the issue of overconfidence, where values close to the ideal boundary were penalized to improve the overall performance. However, in our CTru model, we conducted experiments and found that the use of a penalized term did not impact the deepfake detection performance. This maybe due to the fact that our CTru model already employs dual losses in two spaces, with the same predefined prototypes. In the spherical space, the non-overlapping loss $\mathcal{L}_{\mathrm{S}}$ does not reach 0, which prevents overconfidence in the hyperbolic space.

## 3.5 INFERENCE

Leveraging the acquired features, we devise a novel and highly effective "fakeness" score that considers the model's confidence in hyperbolic space and the cosine similarity in hypersphere space. Our framework exhibits a hybrid nature, wherein the origin holds a unique significance. Viewed from a Euclidean perspective, the local volumes within the Poincaré ball expand exponentially from the origin towards the boundary. The learned embeddings tend to position more generic or ambiguous objects closer to the origin, while situating more specific objects nearer to the boundary. This property aligns with findings presented In Khrulkov et al. (2020). The distance to the origin provides a natural measure of uncertainty that can serve as a reliable indicator of model confidence. Specifically, input images that are familar to the model (pristine images) will be mapped closer to the boundary, while confusing ones such as deepfakes will be closer to the origin. This indicator, denoted as $\mathrm{scr}_{\mathtt{conf}}$, is the first term in Eq. (7). In spherical space, we leverage the standard cosine similarity between the projected input and its corresponding prototypes. This indicator, denoted as $\mathrm{scr}_{\mathtt{alig}}$, is the second term in Eq. (7) (by "alignment", we mean the angle between the embeddings and the predefined prototypes will be aligned to reduce the loss). Formally, given a testing image $\mathbf{I}_{test}$, its "fakeness" score is computed as:

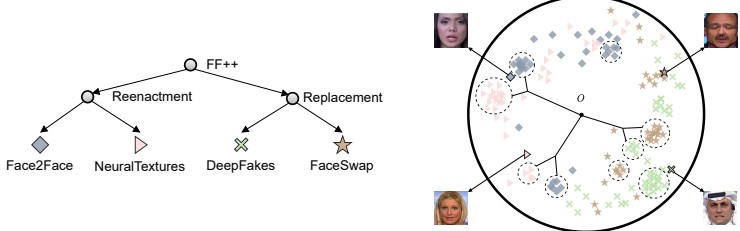

Figure 2: **Left**: Inherent hierarchical structure of different manipulations in FF++ represented as a tree. **Right**: t-SNE projection of embeddings for fake images from the FF++ in the Poincaré ball model. Dotted line circles indicate leaf nodes. CTru effectively captures coarse semantic differences closer to the origin in hyperbolic space (e.g., Reenactment vs. Replacement), with finer-grained distinctions positioned farther away (e.g., Face2Face vs. NeuralTextures).

$$\mathrm{scr}_{\mathtt{fake}}(\mathbf{I}_{test}) = \max_{c \in [1..C]} \underbrace{\mathrm{bus}_{\mathbb{H}}\left(\exp_{\mathbf{0}}(\mathrm{f}_e(\mathbf{I}_{test})), \mathbf{p}_c\right)}_{\text{Confidence } (\mathrm{scr}_{\mathtt{conf}})} \times \underbrace{\left(1 - \mathrm{sim}\left(\mathrm{f}_p^{\mathrm{S}}(\mathrm{f}_e(\mathbf{I}_{test})), \mathbf{p}_c\right)\right)}_{\text{Alignment } (\mathrm{scr}_{\mathtt{alig}})} \quad (7)$$

Our experimental results in Table 1 demonstrate the effectiveness of combining both score in detecting a wide range of deepfake manipulations with high accuracy. We also conducted an ablation study, examining the use of either $\mathrm{scr}_{\mathtt{conf}}$ with $\mathcal{L}_{\mathrm{H}}$ or $\mathrm{scr}_{\mathtt{alig}}$ with $\mathcal{L}_{\mathrm{S}}$ as detailed in Section 4.5

### 3.6 RELATIONSHIP WITH THE CLOSEST PREVIOUS WORKS

**Relation to hyperbolic learning-based prototypes.** Atigh et al. (2021) is the pioneering work that employs the Busemann function as a learning objective. Its main **contribution involves integrating a penalty term**, $\log(1 - |\mathbf{z}_i^{\mathrm{H}}|^2)$, to prevent vectors from approaching the ideal boundary excessively, mitigating the overconfidence issue. The penalty was scaled by $\phi(d)$, a function dependent on the dimension $d$. This function should be linear, specifically $\phi(d) = s \times d$, as the dimension increases, where $s \in [0, 1]$ is the slope parameter.
Considering our loss $\mathcal{L}_H$, we also employ the Busemann function to compute proximities to ideal prototypes but **without the penalty term**. This is equivalent to fixing the slope parameter $s$ to 0. As shown in Equation 7, in $\mathbb{H}^d$, the distance to the ideal prototype, represented by the Busemann function, serves as the confidence metric and thus should not be penalized.
Furthermore, we differ by using evenly distributed ideal prototypes, as detailed in Section 3.1, instead of randomly sampling them from the unit sphere. This allows our points to remain ideal in $\mathbb{H}^d$, while also leveraging optimal properties for the contrastive learning of these points in $\mathbb{S}^d$ using $\mathcal{L}_{\mathrm{S}}$.
**Exploring hierarchy in face-related tasks.** Few recent studies explore hierarchy in face-related tasks. In Gu et al. (2022), deepfake detection is conducted across two hierarchical levels: locally for individual frames and globally using a contrast paradigm applied to video data spanning multiple frames. CTru takes a higher-level abstraction approach that mainly focus on the *inherent hierarchical structure* in the deepfake dataset, as shown in Fig. 2. Guo et al. (2023) identifies the inherent hierarchical structure in deepfake datasets, and classifies manipulated images with multiple labels at different levels in Euclidean space. In contrast, our approach operates without explicit labels, whether hierarchical or single-label, to learn more appropriate feature space.

## 4 EXPERIMENTS

### 4.1 EXPERIMENT SETTINGS

**Datasets and evaluation metrics.** Our model is trained on the *FaceForensics++* (FF++) dataset, as in Tack et al. (2020); Chen et al. (2022a); Zhao et al. (2021). We use only 720 pristine training videos during training and include a maximum of one frame per video when constructing optimization batches. To evaluate CTru in cross-manipulation settings, we assess its performance on the testing portion of the FF++ dataset. This dataset includes videos generated using four generation techniques: Deepfakes (DF) (DeepFakes.), Face2Face (F2F) (Thies et al., 2020), NeuralTextures (NT) (Thies et al., 2019), and FaceSwap (FS) (FaceSwap.). To evaluate the performance of CTru in cross-dataset scenarios, we incorporate three additional datasets: Celeb-DF-v2 (Li et al., 2020b), DeepFake

Table 1: Comparison of CTru and SoTA in the cross-dataset scenario. CTru not only surpasses the second-best method but does so without requiring prior knowledge of face images. * use an additional large-scale dataset (1M or 14M images), ViT has more than five times the parameters as ours.

| Method | Pristine only | Test set - AUC (%) | | | |
| --- | --- | --- | --- | --- | --- |
| | | Celeb-DF (v2) | DFDC | DFDC-p | Average |
| Two-branch (Masi et al., 2020) | | 76.6 | - | - | 76.6 |
| LipForensics (Haliassos et al., 2021) | | 82.4 | 73.5 | - | 77.9 |
| Face X-ray (Li et al., 2020a) | | 79.5 | 65.5 | - | 72.5 |
| SLADD (Chen et al., 2022a) | | 79.7 | - | 76.0 | 77.8 |
| PCL+I2G (Zhao et al., 2021) | ✓ | **90.0** | 67.5 | 74.4 | 77.3 |
| SBI (Shiohara & Yamasaki, 2022) | ✓ | 85.9 | 69.8 | 74.9 | 76.9 |
| OST (Chen et al., 2022b) | | 74.8 | - | 83.3 | 79.1 |
| UIA-ViT (Zhuang et al., 2022) | ✓ | 82.4 | - | 75.8 | 79.1 |
| FTCN-TT (Zheng et al., 2021) | | 86.9 | 74.0 | - | 80.4 |
| LTTD (Guan et al., 2022) | ✓ | 89.3 | - | 80.4 | - |
| IIL (Dong et al., 2023) | ✓ | - | - | 73.8 | - |
| IIL + SBI | ✓ | - | - | 79.6 | - |
| IID (Huang et al., 2023) | ✓ | 82.0 | - | 81.2 | - |
| TALL + EffNet* (Xu et al., 2023) | | 83.4 | 67.1 | - | 75.3 |
| TALL + ViT-B* (Xu et al., 2023) | | 86.6 | 74.1 | - | 80.3 |
| SeeABLE (Larue et al., 2023) | ✓ | 87.3 | 75.9 | 86.3 | 83.2 |
| **CTru** (ours) | ✓ | 89.4 | **77.0** | **87.9** | **84.8** |

Detection Challenge preview (DFDC-p) and DFDC public (DFDC) (Dolhansky et al., 2020). As our evaluation metric, we adopt Area Under the Receiver Operating Characteristic Curve (AUC) as in prior research (Guan et al., 2022; Chen et al., 2022b; Larue et al., 2023).

**Data preprocessing and augmentation.** For datasets without provided face crops, we perform the following preprocessing steps: (1) we first extract 50 frames from each video and apply RetinaNet (Deng et al., 2020) to detect the face regions, and then (2) we use Dlib (King, 2009) to extract the 68-point facial landmark. For each image, the global transformations are first applied with the following operations: random translations, random scaling followed by center cropping, and random shifting of HSV channel values. Then for each patch (we use a $4 \times 4$ grid, $N_{loc} = 16$, details in Section 3.2), we genereated its forged version in both spatial and frequency domains ($N_{typ} = 2$). In the spatial domain, we use random shifting of HSV, RGB channel values, random scaling of the brightness and contrast. In the frequency domain, one of the following operators is used: (1) down-sampling, sharpening/blending filter, and JPEG compression. The hyperparameter values were selected in a way that resulted in visually subtle anomaly without major artifacts, similarly to Shiohara & Yamasaki (2022); Larue et al. (2023), and are detailed in Appendix.

**Training strategy.** We employ the AdamW optimizer with an initial learning rate of 0.002 and no weight decay. The model is trained for 200 epochs using the cosine scheduler without restart, preceded by a linear warmup of 20 epochs. We use a batch size of 64. Images are resized to $299 \times 299$ before being fed into the model. As many other recent detectors (Shiohara & Yamasaki, 2022; Larue et al., 2023), EffNetb4 (Tan & Le, 2019) with 19M parameters was used as backbone by default. We also include a single linear projection on top of the backbone to generate a 100-dimensional vector ($d = 100$). The output of EffNetb4 is a 1794-dimensional vector and in our experiments we obtain the similar performance with $d = 100$ or 200. Training is conducted on two NVIDIA V100 GPUs and requires approximately 16 hours, while during inference, CTru achieves a frame rate of 74fps on a PC with RTX3060.

## 4.2 CROSS-DATASET PERFORMANCE

Table 1 compares the performance of CTru to many SoTA competitors in multiple categories: (1) *pseudo-deepfake* based methods, (2) *video-based techniques*, (3) *transformer-based* methods. All models are trained on FF++ and tested on datasets not seen during training. CTru demonstrates the best overall performance, even though it only uses twenty percent of videos from FF++ (720 pristine videos from the training set). Unlike other competitors, CTru does not require complex multi-head

adversarial training schemes or the inclusion of deepfake examples during training. We do not rely on additional information as used in video-based techniques (CTru operates on a frame-by-frame basis) or employ more powerful transformer backbones. In future work, we intend to expand CTru to encompass video-based techniques, utilizing transformer-like models, with the aim of achieving enhanced performance.

## 4.3 CROSS-MANIPULATION PERFORMANCE

CTru was evaluated on the four manipulation methods of FF++ (Table 2) following the standard evaluation protocol as in Shiohara & Yamasaki (2022). The raw version of FF++ was used for training, and the HQ (compression c23) version was considered for testing.

When videos are compressed, forgery traces can be dampened and become more difficult to detect in Euclidean or spherical space. This is why methods that rely solely on non-negative curvature spaces tend to yield lower results. In contrast, CTru specifically considers local anomalies and treats them in both hyperspherical and hyperbolic spaces, which provides additional advantages and enables us to achieve better performance even when the global forgery traces are less significant. CTru achieved the best performance among all competing detectors.

Table 2: Cross-manipulation evaluation on FF++ HQ. Existing methods (in Table 1) were not all evaluated in this protocol.

| Method | Test set - AUC (%) | | | | |
|---|---|---|---|---|---|
| | DF | F2F | FS | NT | Avg. |
| Face X-ray (Li et al., 2020a) | - | - | - | - | 87.3 |
| SBI (Shiohara & Yamasaki, 2022) | 97.5 | 89.0 | 96.4 | 82.8 | 91.4 |
| OST (Chen et al., 2022b) | - | - | - | - | 98.2 |
| SLADD (Chen et al., 2022a) | - | - | - | - | 98.4 |
| SeeABLE (Larue et al., 2023) | 99.2 | 98.8 | 99.1 | 96.9 | 98.5 |
| **CTru** (ours) | 99.5 | 99.2 | 99.5 | 97.3 | **98.9** |

## 4.4 ROBUSTNESS ANALYSIS

We evaluated CTru on contaminated samples using various perturbation methods by adopting the standard evaluation framework proposed in Jiang et al. (2020), which has been widely used in (Haliassos et al., 2021; Guan et al., 2022; Haliassos et al., 2022). CTru was trained on the FF++ raw dataset, and subsequent inference was carried out on the test split, which had been intentionally degraded using diverse perturbations (Table 3). It is interesting to note that faces are already cropped

Table 3: Robustness evaluation for different corruptions (CS: color saturation, CC: color contrast, BW: block-wise noise, GNC: gaussian noise, GB: gaussian blur, PX: pixelation). * use additional data during training, the direct performance comparison is unfair. F: Frame-based, V: video-based.

| Method | Modality | CS | CC | BW | GNC | GB | PX | Clean | Avg / Drop |
|---|---|---|---|---|---|---|---|---|---|
| Xception (Chollet, 2017) | F | 99.3 | 98.6 | 99.7 | 53.8 | 60.2 | 74.2 | 99.8 | 81.0 / -17.8 |
| SBI (Shiohara & Yamasaki, 2022) | F | 98.1 | 96.5 | 95.3 | 73.5 | 78.3 | 87.9 | 99.9 | 88.3 / -11.6 |
| SeeABLE (Larue et al., 2023) | F | 98.2 | 96.5 | 96.1 | 76.3 | 82.8 | 91.1 | 99.6 | 90.2 / -9.4 |
| LipForensics (Haliassos et al., 2021) | V* | 99.9 | 99.6 | 87.4 | 73.8 | 96.1 | 95.6 | 99.9 | 92.1 / -7.8 |
| LTTD (Guan et al., 2022) | V* | 98.9 | 96.4 | 96.1 | 82.6 | 97.5 | 98.6 | 99.4 | 95.0 / -4.4 |
| RealForensics (Haliassos et al., 2022) | V* | 99.8 | 99.6 | 98.9 | 79.7 | 95.3 | 98.4 | 99.8 | 95.3 / -4.5 |
| **CTru** (ours) | F | 99.6 | 97.3 | 95.2 | 74.2 | 89.9 | 96.8 | 99.7 | 92.2 / -7.5 |

and resized before being input to the models; therefore, resizing is not evaluated as a form of corruption. Among various frame-based methods, including Xception, Face X-ray, and LipForensics, our CTru demonstrates a remarkable level of robustness against perturbations, while achieving better performance. However, for the specific type of Gaussian noise, two other methods, LTTD and RealForensics, outperform CTru. This maybe due to the fact that these models use additional data during training. For instance, LTTD is a *video transformer-based method* which is more complex than ours, while RealForensics trained the model on *both vision and sound data*, where the sound modality remains unaffected by this type of perturbation, resulting in improved performance.

## 4.5 ABLATION STUDY

**Contribution of different losses and scores.** To investigate the impact of $\mathcal{L}_H$ and $\mathcal{L}_S$, we conducted an ablation study in both cross-manipulation (FF++, HQ) and cross-dataset (DFDC) scenarios. We evaluated our method by keeping both losses or removing one of them. We also considered the "guidance" loss $\mathcal{L}_G$ used in Larue et al. (2023). Specifically, $\mathcal{L}_G$ is a specialized loss that relies on $\mathcal{L}_S$ and incorporates prior knowledge about face geometry.

Table 4: Contribution of different losses. SeeABLE and CTru are in the second and third blocks, respectively.

| $\mathcal{L}_{\text{SCL}}$ | $\mathcal{L}_{\text{G}}$ | $\mathcal{L}_{\text{S}}$ | $\mathcal{L}_{\text{H}}$ | scheduler | FF++ | DFDC |
|---|---|---|---|---|---|---|
| ✓ | | | | | 91.6 | 66.8 |
| | ✓ | | | | 51.4 | 51.6 |
| | | ✓ | | | 94.8 | 68.5 |
| | | | ✓ | | 91.3 | 62.3 |
| | ✓ | ✓ | | best fixed $\lambda$ | 97.4 | 74.1 |
| | ✓ | ✓ | | varying $\lambda$ | 98.5 | 75.9 |
| | | ✓ | ✓ | $\lambda = 1$ | 98.9 | 77.0 |

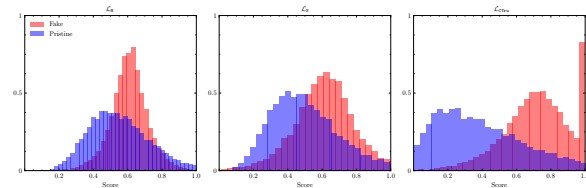

Figure 3: Distributions of the normalized scores of the pristines (blue) and fakes (red) from the DFDC dataset using $\mathcal{L}_{\text{H}}$ (left), $\mathcal{L}_{\text{S}}$ (middle), and $\mathcal{L}_{\text{CTru}}$ (right).

As the "fakeness" score during inference, we utilized only the first term in Eq. (7) when using only $\mathcal{L}_{\text{H}}$. When using $\mathcal{L}_{\text{S}}$ and/or $\mathcal{L}_{\text{G}}$, we did not have $\text{scr}_{\text{conf}}$ in Eq. (7). For a fair comparison, we used the norm of embeddings instead of $\text{scr}_{\text{conf}}$ in Eq. (7), as it has been shown to improve out-of-distribution performance (Tack et al., 2020). Specifically, we replaced $\text{scr}_{\text{conf}}$ in Eq. (7) with $\|\text{f}_e(\mathbf{I}_{test})\|$. When used alone, $\mathcal{L}_{\text{S}}$ and $\mathcal{L}_{\text{H}}$ already performed satisfactorily, while the $\mathcal{L}_{\text{G}}$ loss did not work, likely due to its reliance on prior knowledge and dependence on $\mathcal{L}_{\text{S}}$. When $\mathcal{L}_{\text{S}}$ was combined with $\mathcal{L}_{\text{G}}$, we observed very interesting results. However, these results were dependent on the learning scheduler $\lambda$, which involves hyperparameters that control the balance between the two losses. For instance, when combining $\mathcal{L}_{\text{S}}$ and $\mathcal{L}_{\text{G}}$, several scenarios were considered: (1) $\lambda$ was set to different values (ranging from 0.5 to 5) and the best results were reported, and (2) $\lambda$ was gradually increased or decreased from 0 to 5 or from 5 to 0, respectively. When we used $\mathcal{L}_{\text{S}}$ and $\mathcal{L}_{\text{H}}$ together, we did not require any learning scheduler ($\lambda = 1$), and we achieved new SoTA results. These findings suggest that the two losses are independent yet complementary, simplifying the training process for CTru and making it applicable to a wider range of forensic detection applications without the need for specific geometric constraints as prior knowledge. We further provide visualizations of score distributions for pristine and fake samples obtained from the DFDC dataset using different scores in Figure 3. Using both spaces and scores shows superior separation, resulting in enhanced performance.

**Backbone impact.** Table 5 shows the performance of CTru with several popular CNN backbones. Similar to recent detectors, CTru works best with EffNet-b4 compared to ResNet-50 and Xception.

Table 5: Backbone impact.

| Encoder | FF++ |
|---|---|
| ResNet50 | 95.4 |
| Xception | 95.8 |
| EffNetb4 | 98.9 |

Table 6: Representation learning. *: two-stage with linear probing while others are one-stage.

| Dataset | $\mathcal{L}_{\text{CE}}$ | Atigh et al. (2021) | $\mathcal{L}_{\text{sCL}}$* | $\mathcal{L}_{\text{H}}$ + our points | $\mathcal{L}_{\text{S}}$ | Ours |
|---|---|---|---|---|---|---|
| CIFAR10 | 94.9 | 92.3 | 95.0 | 94.1 | 95.2 | 95.4 |
| CIFAR100 | 76.1 | 65.8 | 76.3 | 68.0 | 76.7 | 77.8 |

**Performance of learnt representation.** We further demonstrate the performance of feature representations learned with different losses using CIFAR10/100 with ResNet-18 backbone in Table 6 (considering larger backbones/datasets for future work). Among methods, $\mathcal{L}_{\text{sCL}}$ is a two-stage approach: it discards the projection layer and necessitates training an additional linear layer using cross-entropy, while others are one-stage. Atigh et al. (2021), "$\mathcal{L}_{\text{H}}$ + our points", $\mathcal{L}_{\text{S}}$, and "Ours" use cosine similarity between the learnt features and prototypes for classification. $\mathcal{L}_{\text{H}}$ with our predefined prototypes outperforms Atigh et al. (2021) (the advantage of the latter compared to CE (cross entropy) becomes more pronounced when the output features have low dimensions). $\mathcal{L}_{\text{S}}$ surpasses $\mathcal{L}_{\text{sCL}}$, and "Ours" performs the best, proving the strength of our learned features in image classification task.

## 5 CONCLUSION

This paper proposes a novel deepfake detector named CTru based on self-supervised anomaly detection on both hyperbolic and hypersphere spaces. During inference, CTru calculates an fakeness score by combining angle-based similarity in spherical space and model confidence in hyperbolic space with Busemann distance. Remarkably, CTru outperforms all the existing SoTA deepfake detectors on various challenging datasets, including both cross-dataset and cross-manipulation scenarios, despite being trained solely on pristine faces.

**Broader impact.** Outstanding results of CTru using only pristine faces highlight the promise of self-supervised anomaly representation learning in spaces with different curvatures for forensic applications, warranting further exploration.

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

APPENDIX

In Section A, we will delve into the implementation details while Section B will offer insights into its relation to the closest literature. We then discuss the exponential growth of hyperbolic spaces and proximity to class (ideal) prototypes in Section C. Finally, in order to make to paper more self-contained, we provide a complete proof for the closed form of Busemann fonction in the Poincaré ball model (Atigh et al., 2021) in appendix D.

## A  IMPLEMENTATION DETAILS

**The position of the forged patch.** We provide a detailed description of the function $\text{loc}(\cdot)$ which was used in the "grid" masking strategy in Section 3.2. This function takes the index $\tilde{y}_{loc} \in [1 \,..\, N_{loc}]$ as input and generates a binary mask $M_{\tilde{y}_{loc}}$ corresponding to the $\tilde{y}_{loc}$-th patch of interest using the grid masking scheme as depicted in Fig. 4 (right).

Given a face image represented as $\mathbf{I} \in [0, 255]^{W \times H \times 3}$, we employ the dlib 68-landmark detector (King, 2009) to extract facial landmarks. Subsequently, a rectangle crop is obtained by determining the bounding box that encapsulates all the landmarks. This rectangular region is further divided into a grid consisting of $N_{loc} = N_{rows} \times N_{cols}$ cells. In order to visualize each region of interest, we employ colored patches as depicted in Fig. 4 (left). The $\tilde{y}_{loc}$-th mask $M_{\tilde{y}_{loc}} \in [0, 1]^{W \times H \times 1}$ is then defined as the intersection between the $\tilde{y}_{loc}$-th region of interest and the convex hull formed by the landmarks.

**Augmentation parameters.** The following hyperparameter values were selected in a way that resulted in visually subtle artifacts, similarly to Shiohara & Yamasaki (2022). Specifically, we applied: (i) random scaling (followed by center cropping) by up to $5\%$, (ii) random shifting of HSV channel values by up to $0.1$, and (iii) random translations of up to $3\%$ and $1.5\%$ of the image width and height, respectively. We also used more fine augmentations: (i) random shifting of HSV channel values by up to $0.3$, (ii) shifting of RGB channel values by up to $20$, (iii) random scaling of the brightness and contrast by a factor of up to $0.1$, or (iv) down-sampling by a factor of $2$ or $4$, (v) a sharpening filter and blending with the original with an $\alpha$ value in the range $[0.2, 0.5]$, (vi) JPEG compression with a quality factor between $30$ and $70$.

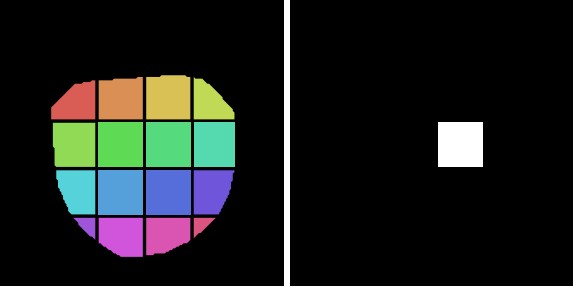

Figure 4: **Left**: illustration of the $4 \times 4$ grid scheme, where the face is divided into a grid of 16 cells (each cell is denoted by a different color). **Right**: The mask $M_{\tilde{y}_{loc}}$ is obtain by setting 1 to the pixels having the same color as the cell at position $\tilde{y}_{loc}$ and 0 otherwise, in this case $\tilde{y}_{loc} = 7$.

## B  RELATIONSHIP WITH THE CLOSEST PREVIOUS WORKS

**Relation to methods that generate pseudo-fakes for deepfake detection.**

Unlike other approaches (Zhao et al., 2021; Chen et al., 2022a; Shiohara & Yamasaki, 2022; Li et al., 2020a) that use data augmentation to create pseudo-fake examples from pristine data and train binary classifiers, we deliberately create localized abnormalities in a controlled manner, without directly aiming to learn the boundary between pristine and pseudo-fake samples.

**Relation to one-class deepfake anomaly detectors.**

While sharing the concept of OC-AD, CTru differs from SeeABLE (Larue et al., 2023) in several ways. Firstly, SeeABLE uses two losses in spherical spaces, whereas CTru uses multiple-curvature spaces, enabling it to capture richer geometric features of human faces. Secondly, the 'guidance' loss in SeeABLE relies on facial geometric constraints as prior knowledge, while CTru uses two independent and complementary losses (Eq. (3) and Eq. (6)) that do not require prior knowledge of the image domain. Thus, CTru is a more versatile model applicable to a broader range of forensic detection tasks. Thirdly, SeeABLE requires a specific learning scheduling for the two losses, while CTru does not, simplifying its training process. CTru outperforms SeeABLE in terms of speed and performance in all evaluated settings. Once trained, CTru achieves a frame rate of 74fps on a PC with RTX 3060, whereas SeeABLE achieves 67fps (for comparison of performance, see Section 4).

## C  THE EXPONENTIAL RELATIONSHIP IN HYPERBOLIC SPACES AND PROXIMITY TO CLASS PROTOTYPES

Our objective is to enhance the proximity of embeddings to the class prototype in two differently-curved spaces. In the hypersphere space, the prototypes $\mathbf{p}_i$ and embeddings $\mathbf{z}_i^S$ lie in the same space. However, in the hyperbolic space, the prototypes are positioned on the boundary of the (open) Poincaré ball and cannot be reached by the hypersphere embedding $\mathbf{z}_i^H$. To accomodate this, we employed an approach that positions our prototypes as ideal points, which correspond to points at infinity from the origin in hyperbolic geometry. We will provide a visualization for this phenomenon within the Poincaré ball model.

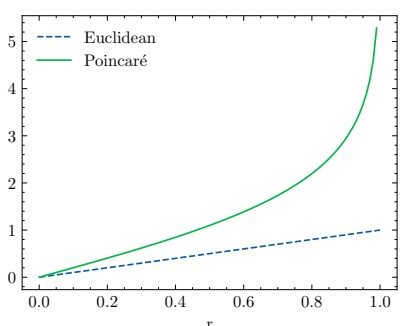

Figure 5: Distance from the origin in the Euclidean and Poincaré ball space as a function of a vector having a norm of $r$.

In Fig. 5, we present a plot depicting the relationship between the Euclidean distance from the origin of a vector $\text{dist}_{\mathbb{R}}\left(r\,\mathbf{x}_E/\|\mathbf{x}_E\|, \mathbf{0}\right)$ of norm $r$ and the Hyperbolic distance Section 3.4 from the origin of its hyperbolic projection $\text{dist}_{\mathbb{H}}\left(\exp_{\mathbf{0}}^{\mathbf{r}}\left(r\,\mathbf{x}_E/\|\mathbf{x}_E\|\right), \mathbf{0}\right)$, across different radius $r \in \mathbb{R}^+$. Specifically, the plot reveals that as $r$ approaches the normalized radius of the ball (i.e., $r = 1$), the distance from the origin exhibits a linear relationship with the vector's norm in Euclidean space. However, in the Poincaré ball space, this relationship follows an exponential pattern. Consequently, as the vector approaches the boundary of the ball, the distances become infinitely large.

## D  COMPLETE PROOF FOR THE CLOSED FORM BUSEMANN FUNCTION IN POINCARÉ BALL MODEL

For the sake of completeness, we also include an adapted version of the proof of Atigh et al. (2021). Starting from the definition of the Busemann function:

$$\text{bus}\left(\mathbf{z}_i^H, \mathbf{p}\right) = \lim_{t\to\infty}\;\left\{\text{dist}_{\mathbb{H}}(\gamma_{\mathbf{p}}(t), \mathbf{z}_i^H) - t\right\}$$

In the Poincaré model, the unit-speed geodesic $\gamma_{\mathbf{p}}(t)$ from the origin towards the ideal point $\mathbf{p}$ is given by:

$$\gamma_{\mathbf{p}}(t) = \mathbf{p}\tanh\left(\frac{t}{2}\right).$$

By substituting this expression into the definition and using the hyperbolic distance as defined in Section 3.4, we can derive the following representation of the Busemann function:

$$\text{bus}_{\mathbb{H}}(\mathbf{p}, \mathbf{z}_i^H) = \lim_{t\to\infty}\left\{\cosh^{-1}(1 + x(t)) - t\right\}, \quad x(t) = 2\frac{\|\mathbf{p}\tanh(\frac{t}{2}) - \mathbf{z}_i^H\|^2}{(1 - \tanh(\frac{t}{2})^2)(1 - \|\mathbf{z}_i^H\|^2)}.$$

By expressing the inverse hyperbolic cosine using the logarithm $\cosh^{-1}(x) = \log(x + \sqrt{x^2 - 1})$, we obtain:

$$\cosh^{-1}(1 + x(t)) = \log\left(1 + x(t) + \sqrt{x(t)^2 + 2x(t)}\right)$$
$$= \log(2x(t) + o(x(t)))$$

where little $o$ represents a lower-order term in Landau's notation. Consequently, we can deduce that:

$$\mathrm{bus}_{\mathbb{H}}(\mathbf{z}_i^{\mathrm{H}}, \mathbf{p}) = \lim_{t \to \infty} \{\log(2x(t) + o(x(t))) - t\}$$
$$= \lim_{t \to \infty} \log\left(2e^{-t}x(t) + e^{-t}o(x(t))\right)$$
$$= \log\left(2 \lim_{t \to \infty} e^{-t}x(t) + 0\right)$$

By utilizing the fact that $\tanh(\frac{t}{2}) = \frac{e^t - 1}{e^t + 1}$, we can conclude that:

$$\lim_{t \to \infty} e^{-t}x(t) = \frac{\|\mathbf{p} - \mathbf{z}_i^{\mathrm{H}}\|^2}{2(1 - \|\mathbf{z}_i^{\mathrm{H}}\|^2)}.$$

The following closed form can be derived, as stated in Eq. (5):

$$\mathrm{bus}_{\mathbb{H}}(\mathbf{z}_i^{\mathrm{H}}, \mathbf{p}) = \log\left(\frac{\|\mathbf{p} - \mathbf{z}_i^{\mathrm{H}}\|^2}{1 - \|\mathbf{z}_i^{\mathrm{H}}\|^2}\right)$$

