# OpenReview forum: "Deepfake Detection with Contrastive Learning in Curved Spaces"
_ICLR.cc/2024/Conference — Submitted to ICLR 2024_

### Official Review · Reviewer_GhyQ · 2023-10-18

**Soundness:** 3 good
**Presentation:** 2 fair
**Contribution:** 2 fair
**Rating:** 5
**Confidence:** 4

**Summary:**

This paper considers the problem of face deepfakes detection. It uses a supervised contrastive learning where a prior set of possible
modifications/alteration  of faces is used as data augmentation. The main novelty of the paper comes from the use of a mixed curvature space
for the embedding, designed as a product of hyperspherical and hyperbolic geometries. Within this geometries, prototypes are defined
as corresponding to the different classes of possible alterations of pristine faces. Then a dissimilarity measure to those mixed-space prototypes
is defined as a combination of a distance over the sphere and a measure of alignment with a hyperbolic prototype thanks to the Busemann
function. A detection score is crafted as a product of similarity in the hyperspherical embedding and a confidence score in the hyperbolical space
defined as the distance to the origin. Thorough experiments are conducted on the FaceForensics++ dataset, and comparisons with SOTA approaches
reveal added value of the mixed-space representation.

**Strengths:**

- Empirical evidences thorough experiments of enhanced detection performances with the proposed method ;
 - although I am not an expert in deepfake detection, the considered SOTA seems relevant and complete

**Weaknesses:**

- the paper combines two well-known strategies (contrastive learning on an hypersphere embedding and Busemann prototypes
from the Ghadimi et al. Neurips paper). The amount of novelties with this respect is low, and one could expect from such a paper
a better justification of the choice of this mixed-curvature space besides ‘the manifold of faces is complicated and non-Euclidean’.
Notably, it is not clear which aspects necessary to deepfake detection is captured by the two geometries
- some details are missing from the experimental part (see my questions below). The ablation study is not fully convincing to me

All in all, and though the proposed approach seems novel and has merits, it seems to me that the paper would be more suited and  impactful in the computer vision community, as far as the novel insight wrt. representation learning are rather limitated.

**Questions:**

- in the experimental section, I did not see the dimensions used fo both embeddings (I may have overlooked). Are they comparable to what is used in  other supervised contrastive learning strategies ? What is the impact of those dimensions on performances ?
- in the ablation study, do you keep the total number of dimensions constant (e.g. if S^100 + H^100 is used, do you compare with a hyperspherical embedding with dimension S^200 ?) I really believe that this question (effectiveness of combination of spherical and hyperbolic geometry) is unsufficiently detailed in the paper)

---

> ### Author Response · Authors · 2023-11-17
> **Reply to Reviewer GhyQ (1/2)**
>
> Thank you for providing constructive reviews. We have addressed each of your questions individually in our response and hope these clarifications contribute to an improved rating. We have revised the paper according to your feedback, marking the changes in magenta. Additionally, we've highlighted in brown the text that was either part of the initial submission or present in the Appendix, which may have been overlooked.
>
> **Question: the paper combines two well-known strategies ... All in all, and though the proposed approach seems novel and has merits, it seems to me that the paper would be more suited and impactful in the computer vision community, as far as the novel insight wrt. representation learning are rather limited.**
>
> **Reply:**
>
> Thank you for your remark and recognition.
>
>
> 1. First, we believe that our approach goes beyond a simple combination of two known learning methods. In relation to existing works, different novelties (new loss terms, new ideal prototypes with improved properties) were considered in each space. Additionally, a new method for generating predefined/ideal prototypes was proposed.
>
> - In hypersphere space (Sec. 3.3), novel terms (the first terms in Eq.3) were incorporated instead of relying solely on the well-known supervised contrastive loss (sCL). This results in improved performance, as demonstrated in Tab.4.  In essence, $L_S$ differs from and outperforms $L_{sCL}$.
>
> - In hyperbolic space (Sec. 3.4), we already clarified the distinctions between our method and that of Atigh et al. (2021) in Appendix (we moved it to the main paper). The main contribution of Atigh et al. (2021) is to integrate a penalized term into the Busemann function to avoid the overconfidence issue. In contrast, we just employ the Busemann function but **without** the penalized term. In our experiments, we find that when using two considered complementary losses, penalizing the distance to the ideal prototype is unnecessary. Moreover, unlike Atigh et al. (2021) that **randomly** samples points from the unit sphere, we use **the predefined evenly distributed points** as ideal prototypes.
>
> - We introduce a novel method to generate evenly distributed vectors (at the end of Sec 3.1) that can be used as both predefined prototypes in sphere space and as ideal prototypes in hyperbolic space.
>
> 2. Second, we have already conducted the experiments to demonstrate the hierarchical relations in deepfake detection datasets. Originally located in the Appendix, we have now integrated them into the main paper. Please refer to Figure 2.
> “Left: Inherent hierarchical structure of different manipulations in FF++ represented as a tree. Right: t-SNE projection of embeddings for fake images from the FF++ in the Poincaré ball model. Dotted line circles indicate leaf nodes. CTru effectively captures coarse semantic differences closer to the origin in hyperbolic space (e.g., Reenactment vs. Replacement), with finer-grained distinctions positioned farther away (e.g., Face2Face vs. NeuralTextures).”
>
> We also developed the explanation of using spaces with different curvatures in page 1.
>
> 3. Third, we also think that the representation learning aspect may have been overlooked. In Fig. 1, given an image I, we compute $z=f_e(I)$ while $f^S_p$ and $f^H_p$ are only two non-parameter projection operators: $z^S=f^S_p(z) = z/norm(z)$; $z^H=f^H_p(z)=exp_0(z)$ (defined in Sec 3.3 and 3.4). We only have a single vector $z$ which conveys the rich information extracted from image I, **not two**. Our approach involves pushing $z^S$ and $z^H$ towards the predefined/ideal prototypes as part of the learning objective for $z$.
>
> - We further evaluate the performance of feature representations learned with different losses using CIFAR10/100 with ResNet-18 backbone in **Tab. 6 in the revised paper** (considering larger backbones/datasets for future work).  $L_H$ with our predefined prototypes outperforms Atigh et al. (2021). $L_S$ surpasses $L_{sCL}$, and ``Ours'' performs the best, proving the strength of our learned features in image classification tasks.
>
> Dataset  | $L_{CE}$  |  Atigh et al. |  $L_{sCL}$  |  $L_{H}$ + our prototypes | $L_{S}$  | Ours
> ---|:---:|:---:|---:|:---:|:---:|:---:
> CIFAR10 | 94.9 | 92.3 | 95.0  | 94.1 | 95.2 | 95.4
> CIFAR100 | 76.1 | 65.8 | 76.3  | 68.0 | 76.7 | 77.8

---

> ### Author Response · Authors · 2023-11-17
> **Reply to Reviewer GhyQ (2/2)**
>
> Dear reviewer,
>
> The initial submission may have contained some ambiguities in certain details. We have thoroughly revised the paper in accordance with your reviews and hope that this prompts you to reconsider your rating.
>
> 1. **Question**. In the experimental section, I did not see the dimensions used for **both** embeddings (I may have overlooked). Are they comparable to what is used in other supervised contrastive learning strategies ? What is the impact of those dimensions on performances?
>
> **Reply:**
> Thank you for your questions.
>
> As mentioned above, $f^S_p$ and $f^H_p$ are only two **non-parameter** projection operators, and we only have **one embedding** $z$ which conveys the rich information extracted from image I, **not two**.
>
> As discussed in Sec 3.3, *supervised contrastive learnings are two-stage approaches*. During inference, they discard the projection layer obtained during training and require training additional linear layer(s) (using cross entropy in the original paper, cross-entropy). In contrast, **our method is one-stage**. Once trained either with $L_H$, $L_S$ or both (using predefined ideal prototypes), we can use the network as is (without removing any layer or training additional layer). Classification is **directly** done by comparing the network’s output with predefined prototypes. In our experiments, we achieved similar performance with dimensions of 100 and 200. Similar findings have been observed in previous representation learning works when the output dimension of the projector is relatively low like 128 or 256. We plan to evaluate the performance of our method with higher output dimensions on larger datasets/backbones. However, it is worth noting that the most challenging deepfake datasets are relatively small compared to large-scale datasets comprising of millions samples.
>
> 2. **Question**. In the ablation study, do you keep the total number of dimensions constant (e.g. if S^100 + H^100 is used, do you compare with a hyperspherical embedding with dimension S^200 ?)...
>
> **Reply:**
> Thank you for your questions.
>
> The dimension of $z^S$ and $z^H$ is identical to that of $z$, **yet at no point during training/inference do we combine the two vectors** $z^S$ and $z^H$. Our approach involves pushing $z^S$ and $z^H$ towards the predefined/ideal prototypes as part of the learning objective for $z$. As mentioned above, in our experiments, we achieved similar performance with relatively low dimensions of 100 and 200. We further demonstrated the effectiveness of our method in image classification tasks in Tab. 6.
>
> We are eager to continue the discussion. Have all your concerns been addressed in our rebuttal, or are there any remaining comments you would like us to consider?

---

> > ### Comment · Reviewer_GhyQ · 2023-11-22
> >
> > Dear Authors
> >
> > Thank you for adressing my concerns and your detailed review. I appreciate the efforts put into this process.
> > Now I understand better why there are no two separate geometric spaces, but only one embedding. I might have been misled by Figure 1.
> > I think one issue remain: you highlight a difference with Atigh et al. (2021) by saying that you have a uniform distribution over the hyperpshere, contrary to a random one. After checking the Atigh et al. (2021) paper, it seems not entirely true, since they use an initialization similar to the one used in the Hyperspherical Prototype Learning paper (from what I understood)
> >
> > I have updated my socre positively, but I still believe that the amount of novlety in the paper is rather moderated.

---

> ### Author Response · Authors · 2023-11-22
>
> Dear reviewer,
>
> Thank you for your thoughtful and constructive feedback. We appreciate your acknowledgment of the efforts we've invested in addressing your concerns and providing a detailed review.
>
> Upon reevaluation, we acknowledge the importance of clarity in understanding the differences between our approach and Hyperspherical Prototype Networks (HPN) [1]. In our method, denoting the set of $K$ class labels as $C = \\{1,..,K\\}$ and the input dimensionality as $L$, we, like HPN, incorporate K prototypes $P = \\{p_{1},...,p_{K}\\}$ within the hypersphere. However, it is crucial to emphasize that our definition of uniformity and the methodology employed for prototype generation diverge from those presented in HPN.
>
> * First, the definition of prototype optimality in [1] differs from ours. They describes their points as *approximately* uniform.
>   We specifically opt for our points to be *evenly distributed*. Formally, in a Euclidean space $R^L$, a set $P$ of $K$ prototypes is said to be evenly distributed only if: $ \\forall\\, i \\neq j, \\;  p_{i} \\cdot p_{j} = -1/(K-1) $.
> > the optimal set of prototypes, $P^*$, is the one where the largest cosine similarity between two class prototypes.$p_i$, $p_j$ from the set is minimized:
> \label{eq:hpn_proto}
> $$
>  P^{*}  = min_{{P}' \\in P} \\bigg( \\max_{(k, l, k\\not=l) \\in C} \\cos(\theta_{(\mathbf{p}'_k,\mathbf{p}'_l)}) \bigg)
> $$
>
> * Second, in [1], prototypes are derived through the gradient descent of a priori points before training. In contrast, our approach generates the set of points through a *closed form solution* that employs the vertices of a $L$-simplex, as elaborated in the supplementary material.
> > To position hyperspherical prototypes prior to network training, we rely on a gradient descent optimization for the loss function of~\\cref{hpn_proto}.
>
> * Finally, our points exhibit the exact property of being *evenly distributed*, which is *contrastive-optimal*. It is noteworthy that contrastive losses attain their minimum once the representations of each class collapse to the vertices of a regular simplex, as demonstrated in [2].
> Additionally, it's important to mention that the vertices of a simplex form an ETF frame and are evenly distributed. We also note that these prototypes are "Busemann" ideal and can therefore be utilized in the Poincaré ball model as prototypes.
>
> Please note that relying solely on the closed form of the simplex results in the same identical set of points, which may be suboptimal due to numerous zeros in the upper triangular part of matrix prototypes. To address this limitation, our paper introduces a three-step generation process. We start with the closed form, add zero padding, apply the Gram-Schmidt process to compute a random basis, and then project the intermediate prototypes onto this basis. This process ensures a *unique* set that adheres to the *evenly distributed* property.
>
> We appreciate your meticulous review of these aspects, and we will ensure that the manuscript accurately reflects these nuances. If you have further specific points or queries related to this comparison, we would be grateful for the opportunity to address them.
>
> Thank you again for your valuable feedback.
>
> [1] Pascal Mettes, Elise van der Pol, and Cees G. M. Snoek. Hyperspherical prototype networks. In
> NeurIPS, 2019.
>
> [2] Florian Graf, Christoph D. Hofer, Marc Niethammer, and Roland Kwitt. Dissecting supervised
> contrastive learning. In ICML, 2021.

---

### Official Review · Reviewer_L6YJ · 2023-10-30

**Soundness:** 3 good
**Presentation:** 2 fair
**Contribution:** 3 good
**Rating:** 6
**Confidence:** 3

**Summary:**

In this paper, the authors propose building facial features by incorporating principles of hyperbolic geometry and using a contrastive loss on a hypersphere to aggregate similar faces, thereby achieving the goal of detecting forged faces. In general, this paper presents a promising approach that contributes to the advancement of deepfake detection.

**Strengths:**

1. An interesting method for constructing facial features.
2. An effective attempt for using contrastive loss to detect deepfakes.

**Weaknesses:**

1. The authors should provide more case studies in the main manuscript, including new features in Section 3 and facial features after clustering using contrastive loss.
2. More analysis on efficiency should be added, such as overall training time, parameters, and convergence steps.
3. It is suggested that the authors consider applying this method to other well-known backbones, such as Xception.

**Questions:**

na

---

> ### Author Response · Authors · 2023-11-17
> **Reply to Reviewer L6YJ**
>
> Thank you for providing constructive reviews. We have addressed each of your questions individually in our response and hope these clarifications contribute to an improved rating. We have revised the paper according to your feedback, marking the changes in magenta. Additionally, we've highlighted in brown the text that was either part of the initial submission or present in the Appendix, which may have been overlooked.
>
> 1. **Question. The authors should provide more case studies in the main manuscript, including new features in Section 3 and facial features after clustering using contrastive loss.**
>
> **Reply:**
> Thank you for your suggestion. Indeed, we have already conducted the experiments to visualize the features after clustering using contrastive loss. **Originally located in the Appendix, we have now integrated them into the main paper.** Please refer to Figure 2.
> ``Left: Inherent hierarchical structure of different manipulations in FF++ represented as a tree. Right: t-SNE projection of embeddings for fake images from the FF++ in the Poincaré ball model. Dotted line circles indicate leaf nodes. CTru effectively captures coarse semantic differences closer to the origin in hyperbolic space (e.g., Reenactment vs. Replacement), with finer-grained distinctions positioned farther away (e.g., Face2Face vs. NeuralTextures).''
>
>
> 2. **Question. More analysis on efficiency should be added, such as overall training time, parameters, and convergence steps.**
>
> **Reply:**
> Thank you for your remarks. Some implementation details (number of training epochs, scheduler, inference time) were already provided but some were in Appendix. We synthesized them in the main paper.
>
> We employ the AdamW optimizer with an initial learning rate of $0.002$ and no weight decay.
> The model is trained for 200 epochs using the cosine scheduler without restart, preceded by a linear warmup of 20 epochs. We use a batch size of 64. Images are resized to $299 \times 299$ before being fed into the model.
> As many other recent detectors like SBI, SLADD, we use EffNetb4 with *19M parameters* as backbone by default. Training is conducted on two NVIDIA V100 GPUs and requires approximately *16 hours* while during inference, CTru achieves a frame rate of 74fps on a PC with RTX3060.
>
> 3. **Question: It is suggested that the authors consider applying this method to other well-known backbones, such as Xception.**
>
> **Reply:**
> Thank you for your suggestion. Xception has been widely utilized in deepfake detection. However, more recent studies, such as SBI, SLADD, have indicated that among CNN backbones, EffNet-b4 has demonstrated superior performance to Xception. In response to your recommendation, we conducted an evaluation of our algorithm using ResNet-50 and Xception. **As shown in Table 5 in the revised version**, our findings align with prior research. Currently, we are exploring Transformer backbones, but please note that they come with a higher parameter count and require pretraining on larger datasets than deepfake datasets.

---

> > ### Comment · Reviewer_L6YJ · 2023-11-21
> > **Concerns are addressed.**
> >
> > Thanks for the response. My concerns are addressed.

---

> > > ### Author Response · Authors · 2023-11-21
> > >
> > > Dear reviewer,
> > > Thank you for your response. Would you kindly consider raising your rating?

---

### Official Review · Reviewer_gKak · 2023-10-30

**Soundness:** 3 good
**Presentation:** 2 fair
**Contribution:** 2 fair
**Rating:** 5
**Confidence:** 4

**Summary:**

This paper proposes to learn deepfake detection representations across multiple-curvature spaces in a self-supervised manner. The detection results combine advantages of both positive and negative curvature spaces. Experimental results validate the effectiveness of the proposed method.

**Strengths:**

1. The proposed model is the first attempt to learn representations across multiple-curvature spaces for deepfake detection.
2. The proposed abnormal face generation method can generate fake faces of many different types.
3. The experimental results show that the proposed model has satisfactory deepfake detection performances.

**Weaknesses:**

The reason to combine both negative and positive curvature representation spaces in deepfake detection is not insightful. This makes the paper merely a combination of existing techniques.

(1) The authors only emphasize that the Euclidean-based distances appear sub-optimal for faces as the complexity and nature of human faces go beyond a basic Euclidean manifold. This explanation is vague and general, thus not convincing.
(2) As I know, using hyperbolic space representations always work well for the tasks with hierachical relation nature. However, the authors fail to explain the inherent hierachical relations in deepfake detection tasks.

**Questions:**

I suggest the authors give more detailed and insightful analysis to explain the motivation of using curved spaces.

---

> ### Author Response · Authors · 2023-11-17
> **Reply to Reviewer gKak (1/2)**
>
> Dear reviewer,
>
> Thank you for providing constructive reviews. We have addressed each of your questions individually in our response and hope these clarifications contribute to an improved rating. We have revised the paper according to your feedback, marking the changes in magenta. Additionally, we've highlighted in brown the text that was either part of the initial submission or present in the Appendix, which may have been overlooked.
>
> **Question: The reason to combine both negative and positive curvature representation spaces in deepfake detection is not insightful. This makes the paper merely a combination of existing techniques.**
>
> **Reply:**
>
> Thank you for your remarks.
>
> 1. First, we believe that our approach goes beyond a simple combination of two known learning methods. In relation to existing works, different novelties (new loss terms, new ideal prototypes with improved properties) were considered in each space. Additionally, a new method for generating predefined/ideal prototypes was proposed.
>
> - In hypersphere space (Sec. 3.3), novel terms (the first terms in Eq.3) were incorporated instead of relying solely on the well-known supervised contrastive loss (sCL). This results in improved performance, as demonstrated in Tab.4.  In essence, $L_S$ differs from and outperforms $L_{sCL}$.
>
> - In hyperbolic space (Sec. 3.4), we already clarified the distinctions between our method and that of Atigh et al. (2021) in Appendix (we moved it to the main paper). The main contribution of Atigh et al. (2021) is to integrate a penalized term into the Busemann function to avoid the overconfidence issue. In contrast, we just employ the Busemann function but **without** the penalized term. In our experiments, we find that when using two considered complementary losses, penalizing the distance to the ideal prototype is unnecessary. Moreover, unlike Atigh et al. (2021) that **randomly** samples points from the unit sphere, we use **the predefined evenly distributed points** as ideal prototypes.
>
> - We introduce a novel method to generate evenly distributed vectors (at the end of Sec 3.1) that can be used as both predefined prototypes in sphere space and as ideal prototypes in hyperbolic space.
>
> 2. Second, we think that the representation learning aspect may have been overlooked. In Fig. 1, given an image I, we compute $z=f_e(I)$ while $f^S_p$ and $f^H_p$ are only two non-parameter projection operators: $z^S=f^S_p(z) = z/norm(z)$; $z^H=f^H_p(z)=exp_0(z)$ (defined in Sec 3.3 and 3.4). We only have a single vector $z$ which conveys the rich information extracted from image I, not two. Our approach involves pushing $z^S$ and $z^H$ towards the predefined/ideal prototypes as part of the learning objective for $z$.
>
>
> - We further evaluate the performance of feature representations learned with different losses using CIFAR10/100 with ResNet-18 backbone in **Tab. 6 in the revised paper** (considering larger backbones/datasets for future work).  with our predefined prototypes outperforms Atigh et al. (2021).  surpasses , and ``Ours'' performs the best, proving the strength of our learned features in image classification tasks.
>
> Dataset  | $L_{CE}$  |  Atigh et al. |  $L_{sCL}$  |  $L_{H}$ + our prototypes | $L_{S}$  | Ours
> ---|:---:|:---:|---:|:---:|:---:|:---:
> CIFAR10 | 94.9 | 92.3 | 95.0  | 94.1 | 95.2 | 95.4
> CIFAR100 | 76.1 | 65.8 | 76.3  | 68.0 | 76.7 | 77.8
>
> - It is worth noting that supervised contrastive learnings are two-stage approaches. During inference, they discard the projection layer obtained during training and require training additional linear layer(s) (using cross entropy in the original paper, cross-entropy). In contrast, our method is one-stage. Once trained either with ,  or both (using predefined ideal prototypes), we can use the network as such (without removing any layer or training additional layer). Classification is directly done by comparing the output of network with predefined prototypes.

---

> > ### Author Response · Authors · 2023-11-17
> > **Typos in Reply to Reviewer gKak (1/2)**
> >
> > There were typos in our previous reply, where some mathematical symbols were not correctly displayed.
> >
> > It should be:
> >
> > $L_H$ with our predefined prototypes outperforms Atigh et al. (2021). $L_S$ surpasses $L_{sCL}$, and ``Ours'' performs the best, proving the strength of our learned features in image classification tasks.

---

> > ### Comment · Reviewer_gKak · 2023-11-22
> >
> > Thanks for your detailed reply.
> >
> > I agree that you have made some improvements beyond a simple combination of known methods. However, these improments, including extra or removed loss terms, or prototype generation method, are not significant enough.
> >
> > In your reply 2/2, The explanation to my first question is still not specific enough. The key point should be "why curved spaces are better for deepfake detection" rather than "why curved spaces are better for human face representation".  The hierarchical relation may answer the first question and figure 2 in the revised version is useful. From this perspective, your logical chain of "deepfake detection is hierarchical --> curved space representation works well in hierarchical relation learning --> we employ curved space representation" is indeed novel. But the novelty is still not enough for ICLR since the first two steps in this logical chain are purely borrowed from previous works and other contributions are not significant.
> >
> > This work is indeed valuable and I suggest the authors to delve deeper into this topic. By now, I have to keep my original rating.

---

> > > ### Author Response · Authors · 2023-11-23
> > >
> > > Dear reviewer,
> > >
> > > We sincerely appreciate your thoughtful feedback and your acknowledgment that ``our work is valuable, novel, and goes beyond a  combination of known methods''. Unfortunately, despite your acknowledgment and our efforts to address all your concerns, the original rating is maintained. We fully understand the rigorous standards of ICLR and the challenges in quantifying assessments of contributions. However, we are inclined to question the characterization of 'our contributions are not significant enough,' as this term seems overly broad.
> > >
> > > We would like to highlight that:
> > >
> > > 1. We believe our approach to learning in hyperbolic space significantly differs from [1], a paper published in NeuRIPS. While we utilize the known Busemann function, similar to their work, we introduce novel ideal prototypes. While the contribution of their paper is "merly" the penalized term, we show that our method does not need this term.
> > >
> > > 2. With our new prototype generation method, supervised contrastive learning can be employed as a one-stage process, departing from its original two-stage form. We present a well-designed approach to **bridge** the two spaces.
> > >
> > > 3. Our method is the **first** deepfake detector that learns a representation in spaces with different curvature.
> > >
> > > 4. Our method **etablishes** new SOTA results, even compared to papers published after the deadline submission of ICLR (please refer to Table 1).
> > >
> > > When comparing to TALL (ICCV’23), using the same EffNet backbone, CTru demonstrates significant superiority with improvements of **+6 and +10 points** on the CDF-v2 and DFDC datasets, respectively. Notably, CTru even surpasses TALL when using Vit-B pretrained on Image-Net21k, comprising 14 million images (**20000 times more data than ours**) and **five times** as many parameters as CTru. Moreover, the utilization of certain large-scale datasets, such as ImageNet-21K as in TALL, *may not be permitted in industrial applications due to licensing restrictions.*
> > >
> > > 5. We have now included additional results in Table 6, demonstrating our method is useful for the general image classification tasks.
> > >
> > > We hope you kindly reconsider raising your rating.
> > >
> > > [1] Atigh et al., "Hyperbolic busemann learning with ideal prototypes", NeuRIPS 2021.
> > >
> > > [2] Xu et al., “TALL: Thumbnail Layout for Deepfake Video Detection”, ICCV 2023.

---

> ### Author Response · Authors · 2023-11-17
> **Reply to Reviewer gKak (2/2)**
>
> **Question. The authors only emphasize that the Euclidean-based distances appear sub-optimal for faces as the complexity and nature of human faces go beyond a basic Euclidean manifold. This explanation is vague and general, thus not convincing.**
>
> **Reply:**
>
> Thank you for your suggestion. We developed the explanation in page 1. ``Unlike the assumptions of a Euclidean manifold, which presumes a flat and linear space, the intricate shapes and curvatures of facial features such as the nose, eyes, and mouth defy adequate description through Euclidean geometry alone. The human face is not a rigid structure; facial expressions are non-linear deformations of facial features (e.g., when a person smiles, the shape of the face undergoes complex transformations). In other words, human faces challenge Euclidean geometry with their dynamic and non-linear features.''
>
>
> **Question. As I know, using hyperbolic space representations always work well for the tasks with hierarchical relation nature. However, the authors fail to explain the inherent hierachical relations in deepfake detection tasks.**
>
> **Reply:**
>
> We have already conducted the experiments to demonstrate the hierarchical relations in deepfake detection datasets. **Originally located in the Appendix, we have now integrated them into the main paper**. Please refer to Figure 2. Left: Inherent hierarchical structure of different manipulations in FF++ represented as a tree. Right: t-SNE projection of embeddings for fake images from the FF++ in the Poincaré ball model. Dotted line circles indicate leaf nodes. CTru effectively captures coarse semantic differences closer to the origin in hyperbolic space (e.g., Reenactment vs. Replacement), with finer-grained distinctions positioned farther away (e.g., Face2Face vs. NeuralTextures)

---

### Official Review · Reviewer_V3Kg · 2023-11-01

**Soundness:** 2 fair
**Presentation:** 2 fair
**Contribution:** 2 fair
**Rating:** 3
**Confidence:** 4

**Summary:**

This paper proposed a framework, CTru, for fakeface detection. The idea is to project face features into different geometric spaces, and combine the projections into a loss function to learn the encoder with contrastive learning. Some experimental results have been shown for demonstration.

**Strengths:**

Slightly better results.

**Weaknesses:**

1. Novelty: The paper integrates several existing techniques widely used in the computer vision community for the application of fake face detection, with no theoretical justification. Why does such an integration work? Why not other ways? This is one of my major concerns as a publication in ICLR, as to me I feel learning nothing from the paper.

2. Writing: I am not clear how Eqs. 2 and 7 are implemented. Eq. 2 is for generating “high” quality fake images, but why is “high” quality? Eq. 7 is for making decisions, but “how”?

3. Experimental results are slightly higher than the approaches that all are before 2023. Not sure if they are state-of-the-art.

**Questions:**

see my comments

---

> ### Author Response · Authors · 2023-11-17
> **Reply to Reviewer V3Kg (1/3)**
>
> Dear reviewer,
>
> Thank you for providing constructive reviews. We have addressed each of your questions individually in our response and hope these clarifications contribute to an improved rating. We have revised the paper according to your feedback, marking the changes in magenta. Additionally, we've highlighted in brown the text that was either part of the initial submission or present in the Appendix, which may have been overlooked.
>
> **Question. Novelty: The paper integrates several existing techniques widely used in the computer vision community for the application of fake face detection, with no theoretical justification. Why does such an integration work? Why not other ways?**
>
> **Reply:**
>
> 1. First, we believe that our approach goes beyond a simple combination of two known learning methods. In relation to existing works, different novelties (new loss terms, new ideal prototypes with improved properties) were considered in each space. Additionally, a new method for generating predefined/ideal prototypes was proposed.
>
> - In hypersphere space (Sec. 3.3), novel terms (the first terms in Eq.3) were incorporated instead of relying solely on the well-known supervised contrastive loss (sCL). This results in improved performance, as demonstrated in Tab.4.  In essence, $L_S$ differs from and outperforms $L_{sCL}$.
>
> - In hyperbolic space (Sec. 3.4), we already clarified the distinctions between our method and that of Atigh et al. (2021) in Appendix (we moved it to the main paper). The main contribution of Atigh et al. (2021) is to integrate a penalized term into the Busemann function to avoid the overconfidence issue.
> In contrast, we just employ the Busemann function but **without** the penalized term. In our experiments, we find that when using two considered complementary losses, penalizing the distance to the ideal prototype is unnecessary. Moreover, unlike Atigh et al. (2021) that **randomly** samples points from the unit sphere, we use **predefined evenly distributed points** as ideal prototypes.
>
> - We introduce a novel method to generate evenly distributed vectors (at the end of Sec 3.1) that can be used as both predefined prototypes in sphere space and as ideal prototypes in hyperbolic space.
>
> 2. Second, we think that the representation learning aspect may have been overlooked. In Fig. 1, given an image I, we compute $z=f_e(I)$ while $f^S_p$ and $f^H_p$ are only two non-parameter projection operators: $z^S=f^S_p(z) = z/norm(z); z^H=f^H_p(z)=exp_0(z)$ (defined in Sec 3.3 and 3.4). We only have a single vector $z$ which conveys the rich information extracted from image I, not two. Our approach involves pushing $z^S$ and $z^H$ towards the predefined/ideal prototypes as part of the learning objective for $z$.
>
> - We further evaluate the performance of feature representations learned with different losses using CIFAR10/100 with ResNet-18 backbone in **Tab. 6 in the revised paper** (considering larger backbones/datasets for future work). $L_H$ with our predefined prototypes outperforms Atigh et al. (2021). $L_S$ surpasses $L_{sCL}$, and ``Ours'' performs the best, proving the strength of our learned features in image classification tasks.
>
> Dataset  | $L_{CE}$  |  Atigh et al. | $L_{sCL}$   | $L_{H}$ + our prototypes | $L_{S}$ | Ours
> ---|:---:|---:|---:|---:|---:|---:
> CIFAR10 | 94.9 | 92.3 | 95.0  | 94.1 | 95.2 | 95.4
> CIFAR100 | 76.1 | 65.8 | 76.3  | 68.0 | 76.7 | 77.8
>
> - It is worth noting that supervised contrastive learnings are two-stage approaches. During inference, they discard the projection layer obtained during training and require training additional linear layer(s) (using cross entropy in the original paper, cross-entropy). In contrast, our method is one-stage. Once trained either with $L_H$, $L_S$ or both (using predefined ideal prototypes), we can use the network as such (without removing any layer or training additional layer). Classification is directly done by comparing the output of network with predefined prototypes.

---

> ### Author Response · Authors · 2023-11-17
> **Reply to Reviewer V3Kg (2/3)**
>
> **Question. Writing: I am not clear how Eqs. 2 and 7 are implemented. Eq. 2 is for generating “high” quality fake images, but why is “high” quality? Eq. 7 is for making decisions, but “how”?**
>
> **Reply:**
>
> Thank you for your questions.
> - Concerning Eq. 2, we generated our synthetic images using a variety of augmentation techniques, akin to those utilized in contemporary deepfake detectors like [SBI] but our approach introduces only localized artifacts to the images. Due to the space limit, the specific augmentations were detailed in the Appendix and cross-referenced in the main paper. By employing an array of sophisticated augmentation techniques—spanning spatial and frequency transformations, scaling, contrast adjustments, down-sampling, sharpening filters, blending, and more—we effectively simulate diverse fake artifacts.
>
> **In Appendix**. “The following hyperparameter values were selected in a way that resulted in visually subtle artifacts, similarly to Shiohara & Yamasaki (2022). Specifically, we applied to “source”: (i) random scaling (followed by center cropping) by up to 5%, (ii) random shifting of HSV channel values by up to 0.1, and (iii) random translations of up to 3% and 1.5% of the image width and height, respectively. We also used more fine augmentations to “target”: (i) random shifting of HSV channel values by up to 0.3, (ii) shifting of RGB channel values by up to 20, (iii) random scaling of the brightness and contrast by a factor of up to 0.1, or (iv) down-sampling by a factor of 2 or 4, (v) a sharpening filter and blending with the original with an α value in the range [0.2, 0.5], (vi) JPEG compression with a quality factor between 30 and 70.”
>
> - Concerning the use of fakeness score for detecting fake during inference, we follow the standard evaluation protocol for deepfake detection and report the Area Under the Curve (AUC) as the performance metric. We consider the Receiver Operating Characteristic (ROC) curve of the validation split of the FF++ dataset and select the threshold that yields the highest AUC. This threshold remains fixed when evaluating the testing dataset.

---

> ### Author Response · Authors · 2023-11-17
> **Reply to Reviewer V3Kg (3/3)**
>
> **Question. 3. Experimental results are slightly higher than the approaches that all are before 2023. Not sure if they are state-of-the-art.**
>
> **Reply:**
>
> Thank you for your questions.
>
> - At the time of submitting the CTru paper, **we had already carefully verified** the SOTA results on the GitHub
> https://github.com/Daisy-Zhang/Awesome-Deepfakes-Detection which gathers and updates almost all the deepfake papers from CVPR, ICCV, ICLR, NeuRIPS, and so on. In our initial submission, we already included the best known results so far coming from SeeABLE (at that time, the paper was just on arXiv and we now update its official publisher ICCV’23).
>
> - For a clearer comparison, we have now  incorporated in the revised version the results of other recent methods from CVPR’23, ICCV’23 papers in Tab. 1 (please note that those ICCV’23 papers were published after the submission deadline of ICLR 2024).
>
>
> Method     |       Celeb-DF-v2   |       DFDC	|      DFDC-p      |    Note
> ---|:---:|:---:|:---:|:---:
> IIL - CVPR’23 [1]'s Tab.3            |      			     -*	         |           -	   |      73.8 | -
> IIL + SBI (CVPR’23 + CVPR’22 Oral) [1]'s Tab.6       |    	     -*        |                      -	 |           79.6     |    Combine 2 methods
> IID (CVPR’23) [2]'s Tab.2                      |              		    82.0      |                      -	 |           81.2       |  -
> TALL + Swin (ICCV’23) [3]'s Tab.2       |                                  90.7      |                    76.8    |            -     |     x5 paras + 14M images
> TALL + ViT-B (ICCV’23) [3]'s Tab.2  	 |  		    86.6	 |  	     74.1	    |          -	 |       x5 paras + 14M images
> TALL + EffNet (ICCV’23) [3]'s Tab.2        |                              83.4 	 |  	     67.1	      |        -    |    + 1M images
> SeeABLE EffNet (ICCV’23) 		      |               87.3	 |  	     75.9  	  |         86.3   |
> CTru   (EffNet)    				       |              89.4	 |  	     77.0	   |        87.9  |
>
> *: IIL ([1]-Tab. 3) was evaluated on Celeb-DF-v1 which is older and easier than v2.
>
> - Note that TALL [3] requires an additional large-scale dataset to pre-train its backbones (1M and 14M images when using CNN EffNet and Transformer Swin backbones, respectively), while CTru is trained from scratch with only 720 images. It is clear that CTru’s results are SOTA. We believe that our improvement is **not slight**. In comparison with SeeABLE, CTru consistently outperforms by a margin of nearly **2 points**, while maintaining a **faster and more generic** approach (without the requirement of a specific training scheduler and prior knowledge of the face, as mentioned in the paper).
> - When comparing to TALL (ICCV’23), using the same EffNet backbone, CTru demonstrates significant superiority with improvements of **+6 and +10 points** on the CDF-v2 and DFDC datasets, respectively. Notably, CTru even surpasses TALL when *using Vit-B pretrained on Image-Net21k, comprising 14 million images and five times* as many parameters as CTru.
>
> [1] Dong et al., “Implicit Identity Leakage: The Stumbling Block to Improving Deepfake Detection Generalization, CVPR 2023.
>
> [2] Huang et al., “Implicit Identity Driven Deepfake Face Swapping Detection”, CVPR 2023
>
> [3] Xu et al., “TALL: Thumbnail Layout for Deepfake Video Detection”, ICCV 2023

---

> ### Author Response · Authors · 2023-11-17
> **Reply to Reviewer V3Kg**
>
> We would also like to emphasize that the two losses in two spaces are connected through the utilization of the same predefined/ideal prototypes.
>
> We are eager to continue the discussion. Have all your concerns been addressed in our rebuttal, or are there any remaining comments you would like us to consider?

---

### Author Response · Authors · 2023-11-17
**General Reply**

Dear PC, AC and all reviewers,

We appreciate the valuable comments from the four reviewers that help improve the completeness of our submission. We have revised the paper according to their feedback, marking the changes in magenta. Additionally, we've highlighted in brown the text that was either part of the initial submission or present in the Appendix, which may have been overlooked by the reviewers.

Reviewers have expressed concerns regarding 1) the novelty and 2) the positioning of our method in relation to representation learning.
We would like to briefly emphasize the novelties and positioning of CTru as follows (more details can be found in the individual replies to each reviewer):

1. First, we believe that our approach goes beyond a simple combination of two known learning methods. In relation to existing works, different novelties (new loss terms, new ideal prototypes with improved properties) were considered in each space. Additionally, a new method for generating predefined/ideal prototypes was proposed.

2. Second, we think that the representation learning aspect may have been overlooked. In our Fig. 1, $f^S_p$ and $f^H_p$ are only two non-parameter projection operators: $z^S=f^S_p(z) = z/norm(z)$; $z^H=f^H_p(z)=exp_0(z)$ (defined in Sec 3.3 and 3.4). We only have **one embedding $z$, not two**. Our approach involves pushing $z^S$ and $z^H$ towards the predefined/ideal prototypes as part of the learning objective for $z$.

- To reply to the reviewers’ suggestion, we further evaluate the performance of feature representations learned with different losses using CIFAR10/100 with ResNet-18 backbone in Tab. 6 in the revised paper. $L_H$ with our predefined prototypes outperforms Atigh et al. (2021). $L_S$ surpasses $L_{sCL}$, and ``Ours'' performs the best, proving the strength of our learned features in image classification tasks.

We are enthusiastic about continuing the discussion.

---

### Meta-Review · Area_Chair_byp2 · 2023-12-07

**Metareview:**

The paper presents an integration of hyperbolic and hypersphere spaces in deepfake detection, aiming to provide a novel approach in the field. Reviewers expressed concerns about the novelty and theoretical justification of this integration, with some considering it as a mere combination of existing techniques. The authors responded by highlighting the unique aspects of their approach, such as the novel method for generating predefined prototypes and the use of supervised contrastive learning in a one-stage process. Despite these assertions, the reviews suggest that the paper lacks a clear and convincing explanation of the inherent hierarchical relations in deepfake detection tasks and the specific aspects necessary for this detection captured by the two geometries. The experimental results do not convincingly establish state-of-the-art performance.

**Justification For Why Not Higher Score:**

The main reason for not awarding a higher score lies in the paper's inability to convincingly justify the theoretical underpinnings of its approach and its significance in advancing the field of deepfake detection. The integration of existing techniques, while potentially useful, is not sufficiently substantiated with theoretical or empirical evidence to demonstrate a clear advancement over current methods. The reviewers pointed out the vagueness in explaining the relevance of mixed-curvature space and the lack of clarity in the implementation details. Furthermore, the experimental results, though marginally better, do not significantly outperform existing methods to warrant a higher evaluation score.

**Justification For Why Not Lower Score:**

NA

---

### Decision · Program_Chairs · 2024-01-16

Reject